# Severe Microbial Keratitis in Virgin and Transplanted Cornea—Probability of Visual Acuity Improvement

**DOI:** 10.3390/jcm14010124

**Published:** 2024-12-28

**Authors:** Joanna Przybek-Skrzypecka, Małgorzata Ryk-Adamska, Alina Szewczuk, Janusz Skrzypecki, Justyna Izdebska, Monika Udziela, Anna Rypniewska, Leejee H. Suh, Jacek P. Szaflik

**Affiliations:** 1Department of Ophthalmology, Medical University of Warsaw, Sierakowskiego 13, 01-756 Warsaw, Polandjustyna.izdebska@wum.edu.pl (J.I.); monika.udziela@wum.edu.pl (M.U.); jacek.szaflik@wum.edu.pl (J.P.S.); 2SPKSO Ophthalmic University Hospital Warsaw, 03-709 Warsaw, Poland; alina.yarashevich@gmail.com (A.S.); arypniewska@gmail.com (A.R.); 3Department of Experimental Physiology and Pathophysiology, Medical University of Warsaw, 01-756 Warsaw, Poland; janusz.skrzypecki@wum.edu.pl; 4Cornea & Refractive Surgery, Edward S. Harkness Eye Institute, Columbia University Irving Medical Center, New York, NY 10032, USA; lhs2118@cumc.columbia.edu

**Keywords:** keratitis, corneal ulcers, legal blindness, therapeutic corneal transplant

## Abstract

**Purpose:** To evaluate visual acuity improvement and identify contributing factors in patients with severe keratitis affecting both virgin and transplanted corneas, treated at a hospital. **Methods:** A retrospective analysis was conducted on 497 patients with unilateral corneal ulcers treated at a tertiary referral center between 2008 and 2023. Data included distance (BCVA) and near best-corrected visual acuity at initial presentation and at discharge, treatments before hospital admission, demographic details, risk factors, clinical signs and symptoms, ancillary test results, and management strategies. Patients were categorized into two groups: Group A (naïve corneal ulcers, 379 patients) and Group B (post-keratoplasty infectious keratitis, 118 patients). Additional analysis focused on patients with presenting visual acuity of at least 1.0 logMAR (≤5/50 Snellen charts = legal blindness) to predict final visual outcomes. **Results:** The median BCVA at presentation for the entire cohort was 1.9 logMAR, advancing to 1.5 logMAR at discharge (*p* < 0.001). At least one line improvement in BCVA was observed in 47% of patients (52% of naïve cornea and 33% of transplanted cornea patients). Significantly worse results were observed in Group B were observed for BCVA at presentation, BCVA improvement, and distance and near vision improvement. Among patients with legal blindness at presentation, vision status improved for 52/379 (14%) in Group A and 6/118 (5%) in Group B during hospital admission (*p* < 0.001), while 67% of the cohort was discharged with VA equal or worse than 5/50. The average hospital stay was 9 days. Near visual acuity got better in 23% of patients (27% in Group A vs. 9% in Group B). A multivariate regression model showed that older age and worse distance BCVA on admission were independent negative predictors of improvement (*p* < 0.001, *p* < 0.001, respectively) while midperiphery ulcers were associated with better visual outcomes. **Conclusions:** Hospital admission leads to BCVA improvement in 47% of the patients with severe corneal ulcer, though the prognosis is significantly worse for those with post-keratoplasty microbial keratitis. At discharge, 67% of patients remained at the legal blindness level. Older age and lower BCVA at first presentation are associated with worse prognosis, while ulcers located in the corneal midperiphery are linked to better visual outcomes.

## 1. Introduction

Keratitis is a serious ocular condition that can lead to eye loss in severe cases [1,2,3]. There is evidence suggesting that keratitis in transplanted corneas may pose an even higher risk, though available data remain limited [4]. Notably, corneal keratitis accounts for approximately one in fifty ocular emergency department consultations [5]. Severe cases of corneal keratitis (defined as one of the following: centrally located ulcer with a diameter of at least 2 mm, presence of hypopyon, complete corneal opacification, an ulcer at any location with a diameter of at least 3 mm and reduced visual acuity, keratitis affecting a transplanted cornea), are considered a reason for inpatient admission. However, this recommendation is based on low-certainty evidence due to the limited data available on the outcomes of severe keratitis following admission. Given the limited number of inpatient beds in modern ophthalmology departments as well as the high cost of hospitalization, it is crucial to carefully assess and prioritize conditions that would benefit most from admission [5,6]. Although direct evidence in ophthalmology is yet to be established, other direct cost–benefit relationships have been shown for other conditions, namely myocardial infarction and severe pneumonia [7,8,9,10]. 

Our study aims to assess the likelihood of visual acuity improvement and identify contributing factors in hospitalized patients with severe microbial keratitis. Additionally, we compare outcomes between patients with naïve corneas and those with transplanted corneas. To the best of our knowledge, this is the first large cohort study to evaluate and compare the effects of hospital treatment for severe corneal ulcers in both virgin and transplanted corneas.

## 2. Methods

A retrospective analysis was conducted on 517 adults diagnosed with severe microbial keratitis, identified using the International Classification of Diseases (ICD-10) code H16.1, and admitted to the tertiary eye center at the Department of Ophthalmology, Medical University of Warsaw, Poland, between 2008 and 2023. After excluding 20 cases initially coded as microbial keratitis but later diagnosed as non-infectious or immunological ulcers, the final analysis included 497 cases.

### 2.1. Definitions

Microbial keratitis was defined as either microbial growth in corneal cultures or clinical improvement following antimicrobial treatment.

Acanthamoeba keratitis was defined based on characteristic features observed in in vivo confocal microscopy (IVCM) or clinical improvement after initiating anti-amoebal treatment.

Severe keratitis was defined by at least one of the following criteria and fulfilled criteria for hospital admission:(a)A centrally located ulcer with a diameter of at least 2 mm.(b)Presence of hypopyon.(c)Complete corneal opacification.(d)An ulcer at any location with a diameter of at least 3 mm and reduced visual acuity.(e)Keratitis affecting a transplanted cornea.

The cohort was divided into two subgroups:

Group A: Keratitis in naïve (virgin) corneas.

Group B: Keratitis in transplanted corneas (post-keratoplasty microbial keratitis, PKMK).

The study was approved by the Institutional Review Board of the Medical University of Warsaw (approval number: 292/2023) and adhered to the tenets of the Declaration of Helsinki.

### 2.2. Data Collection

The following data was collected: demographics, including sex and age, season of admission, duration of hospitalization, past ocular history, and general medical history.

Risk factors for keratitis were assessed, such as contact lens use, history of ocular trauma, prior treatment, time from symptom onset to ophthalmology visit, and corneal status, categorized as naïve or previously transplanted. The number of infiltrates was documented as single or multiple, while the location of the infection within the cornea was classified as central, defined as the central 5-mm diameter; mid-peripheral, within the 5–8 mm-diameter arc; peripheral, beyond the 8-mm diameter; or involving the entire cornea.

Best-corrected distance visual acuity (BCVA) and near visual acuity were measured at both initial presentation and discharge using logarithm of the Minimum Angle of Resolution (logMAR) units and/or Snellen charts. Extremely low VA was categorized as hand movement corresponding to 2.3 logMAR, light perception to 2.7 logMAR, and no light perception as 3.0 logMAR. Lens status was recorded to control for potential bias from cataract presence. According to the World Health Organization, visual acuity equal to or greater than logMAR 1, corresponding to Snellen ≤ 5/50, was defined as legal blindness, while logMAR between 0.5 and 0.99, equivalent to Snellen 5/16–5/50, indicated severe visual impairment. Data on surgical interventions, including keratoplasty, amniotic membrane transplantation, and evisceration, were also extracted from the medical records.

Empirical therapy was investigated and divided into topical treatments, which included antibacterial monotherapy or polytherapy, antifungal, antiviral, and antiamoebic agents, glucocorticosteroids, disinfectants, cycloplegics, and systemic therapy, which comprised antibiotics, antivirals, and glucocorticosteroids.

### 2.3. Ancillary Testing

Corneal scrapings were collected upon admission. When available, conjunctival swabs, contact lenses, and contact lens storage boxes were also inspected. Samples were cultured using standard media, including blood agar, chocolate agar, and Sabouraud agar. The corneal culture results were categorized into four groups: Gram-positive bacteria, Gram-negative bacteria, fungi, and mixed infections.

“Organism growth” was defined as a positive culture or microbial growth from an anterior chamber tap or conjunctival swab, excluding the natural ocular surface microbiome. In a few cases from 2008 and 2009, the specific tissue source was unclear due to incomplete documentation. Mixed infections were defined as the presence of two types of microorganisms: either a combination of Gram-positive and Gram-negative bacteria or a bacterial-fungal co-infection.

In Vivo Confocal Microscopy was performed using the Confoscan 4 (Nidek Technologies, Gamagori, Japan) when Acanthamoeba or fungal infections were suspected. At our center, Acanthamoeba keratitis is diagnosed clinically and with IVCM, as Escherichia coli-enriched plates required for confirmation are unavailable. Notably, polymerase chain reaction (PCR) method for any microorganism detection was unobtainable at our center in the studied period of time.

### 2.4. Statistical Methods

The analysis was carried out using R statistical software, version 4.0.2. Data was presented with n (% of group) for nominal variables and as mean ± SD or median (Q1; Q3) for continuous variables, depending on distribution normality. Verification of normality was achieved with Shapiro–Wilk, and homogeneity of variance was assessed with Bartlett’s test. For normally distributed data with equal variances, the means were compared using either the Student’s *t*-test or ANOVA for independent variables. For the remaining parameters, appropriate non-parametric statistical methods were selected. The Mann–Whitney U test was used to compare numerical variables between two groups of observations, and the Kruskal–Wallis test with post-hoc Dunn test and Bonferroni correction was used for at least three groups. In order to test the relationship between categorical variables, the Fisher test or the chi-square test were used. Effect sizes were calculated to complement *p*-values and provide a measure of the magnitude of observed differences or associations (to address inequality in number of the subgroup members). The Tukey post-hoc test was used significantly in results of ANOVA analysis. Multiple ANCOVA regression models were used to examine the relationship between the dependent variable and explanatory variables for final visual acuity of legal blindness. In each case, we evaluated the feasibility of a multiple linear regression model as well as the model fit described by the coefficient of determination R^2^, which takes values between 0 and 1 (greater values suggest that our model explains more variability of the dependent variable), adjusted R^2^, and AIC (Akaike information criterion). The level of statistical significance was set to *p* = 0.05.

## 3. Results

### 3.1. Demographics

The analysis included 497 patients admitted to a tertiary eye hospital with a clinical diagnosis of infectious corneal ulcer. Of these, 55.7% were female. The median age was 61. The study included 379 patients with keratitis in naïve corneas (Group A) and 118 post-keratoplasty microbial keratitis patients (Group B). There were no significant demographic differences between the two groups. Seasonal variations were noted, with a higher incidence of keratitis cases occurring in the summer and spring. Table 1 presents the demographic characteristics in detail.

### 3.2. Visual Acuity

The median visual acuity on admission was 1.9 logMAR (Q1:Q3 = 0.9–2.3) for the entire cohort, with a significant difference between the groups (Group A: 1.9 logMAR, Group B: 2.3 logMAR, *p* < 0.001). An improvement of at least one Snellen line was observed in 52% of patients in Group A and 33% in Group B (*p* < 0.001). The influence of cataracts was excluded as no significant difference in lens status was noted (*p* = 0.21). Legal blindness status altered for 11.7% of the cohort from admission to discharge (71% vs. 57% in Group A and 86% vs. 80% in Group B), with significant improvement only in the naïve keratitis group. Of note, median BCVA among legal blindness patients was 2.3 logMAR for the whole cohort (Q1–Q3 1.6–2.3), Group A (Q1–Q3 1.6–2.3), and Group B (Q1–Q3 1.9–2.3) (*p* = 0.19), and shifted to median 1.9 logMAR for the entire group. Any level of near vision was observed in 134 persons (27% of 497) on admission and 173 (35% of 497) at discharge. Near vision improved in 27% of patients in Group A and 9% in Group B (*p* < 0.001). Table 2 presents details of visual acuity parameters and their shift in the studied period of time. Furthermore, additional analysis based on visual acuity levels: ((a) logMAR ≥ 1, (b) logMAR = 0.5–0.99, (c) logMAR < 0.5) was performed. Of note, there were statistically significant difference for age, CL use, corneal part affected (central, midperiphery, peripheral), mono/multifocal involvement, active uveitis at presentation, glaucoma history, therapeutic corneal transplant, and antifungal treatment. The median age for ((a) logMAR ≥ 1 was 67 years, (b) logMAR = 0.5–0.99–51 years and (c) logMAR < 0.5 = 35 years, *p* < 0.001). CL use prevalence increased with initial BCVA (9% in logMAR ≥ 1 group, 25% in logMAR = 0.5–0.99 and 33% in logMAR < 0.5 group).

### 3.3. Risk Factors and Clinical Characteristics

Contact lens wearers constituted 15% of the cohort, with a significant difference between Group A and Group B (19% vs. 1%, *p* < 0.001). Traumatic events occurred in 56 patients, with 14% in Group A and 3% in Group B (*p* < 0.001). The median time from symptom onset to hospital admission was 9 days for the entire cohort, with Group B patients tending to seek specialist care sooner (a median of 7 days, Q1:Q3 = 3–14 days) than Group A patients (a median of 9 days, Q1:Q3 = 3–30 days). More patients in Group B received prior treatment from local ophthalmologists (57% in Group B vs. 52% in Group A). There were no significant differences in uveitis status between the groups (15% vs. 10%), but a significant difference was noted for glaucoma status (12% in Group A vs. 39% in Group B, *p* < 0.001). Table 3 depicts further clinical characteristics, including the percentage of patients with a certain number and specific locations of infiltrates showing no statistically significant difference between the subgroups.

### 3.4. Ancillary Tests

Corneal scrapes were performed in 163 patients. Positive corneal culture occurs in 103 patients from the Group B (72%) and 11 patients in the Group B (55%) (*p* < 0.001, effect size 0.118). In Group A, we obtained 79 mono-organism growth and 24 multi-bacterial/fungal growth. Any microorganism growth (derived either from scrapes, anterior chamber tap, or swabs other than natural ocular surface microbiome) was detected in 84% of the Group A. Microbial growth was proven in 32% for the whole of Group A and 22% in Group B (*p* = 0.041). Gram-positive bacteria dominated in both subgroups (50% in Group A and 63% in Group B, *p* = 0.37) followed by mixed infections (Group A: 23%, Group B: 29%, *p* = 0.70). Almost 10% difference in occurrence was noted for both Gram-negative and fungal infection. Notably, for the whole cohort the percentage of Gram-positive cultures was higher for the elderly group (minimum 65 years old) than in the younger (<65 years old) population (62% vs. 46%, respectively, *p* = 0.09). Confocal imaging was obtained more often in the naïve cornea group than PKMK group (13% vs. 5%, *p* = 0.028). Table 4 depicts diagnostic details of the subgroups.

### 3.5. Treatment

Empirical antibiotic therapy was administered in 94% of patients, with polytherapy being more common than monotherapy (61% in Group A vs. 73% in Group B, *p* = 0.0285). Polytherapy was associated with a higher probability of visual acuity improvement in the univariate regression model (*p* = 0.019). Fungal empirical treatment was introduced in 53% of patients with no statistically significant difference between the groups. However, the univariate regression model provided better visual outcomes in the subgroup of patients with ani-fungal treatment started at the admission. Empirical treatment for Acanthamoeba keratitis was more commonly introduced in Group A (twenty-five vs. one case in Group B, *p* = 0.015). Cycloplegics were applied more frequently in Group A (71% vs. 53%, *p* < 0.001). Surgical intervention was performed on over 40% of the cohort (therapeutic corneal transplant in 19% of patients and amniotic membrane transplantation in 22%), with no significant difference between the groups. Two patients in the whole cohort required evisceration, both in the naïve cornea group. Table 5 demonstrates details of varied modes of treatment in our study.

Table 6 presents the results of the univariate regression model for visual acuity improvement (defined as at least one line on Snellen charts) for the whole cohort. It occurred that: longer hospitalization time, better BCVA at presentation, CL use, midperipheral location, antibacterial polytherapy treatment, and antifungal treatment were statistically significant for better visual outcomes. On the contrary, older age, history of glaucoma, and post-keratoplasty corneal ulcer represent negative predictive factors.

Furthermore, a multivariate regression model was used to identify factors associated with visual acuity enhancement (at least one line on Snellen charts). Low initial visual acuity and older age were strong negative prognostic factors (*p* < 0.001 for both). Midperipheral ulcer location was a significant positive factor for logMAR reduction (*p* = 0.049). Table 7 depicts the analyzed factors in an ANCOVA model.

## 4. Discussion

Our study demonstrates that inpatient treatment significantly improves visual acuity in severe keratitis affecting both virgin and post-keratoplasty corneas, though outcomes were notably worse in the post-keratoplasty microbial keratitis group. BCVA improved by at least one line in 52% of patients with naïve keratitis and 33% of those with PKMK. Median BCVA on admission was 1.9 logMAR (counting fingers) for virgin corneas and 2.3 logMAR (hand movement) for PKMK, improving to 1.3 logMAR (2.5/50 Snellen chart) and 1.9 logMAR (counting fingers), respectively, at discharge. Legal blindness status improved in 14% of patients with naïve keratitis but only in 5% of the PKMK group. Near visual acuity was detectable in 27% of patients at admission and in 35% at discharge, with advancement seen in 27% of the virgin cornea group and just 8% in the PKMK group (*p* < 0.001).

Legal blindness status remains in 67% patients of the entire cohort at discharge.

Low initial VA and older age are widely recognized as unfavorable prognostic factors for microbial keratitis treatment [11,12,13,14,15]. Few studies, however, have measured VA improvements following intensive inpatient treatment or directly compared outcomes between severe keratitis in virgin and transplanted corneas within a single cohort. Unlike prior research, our study included all visual acuity levels, including hand movement (2.3 logMAR), light perception (2.7 logMAR), and no light perception (3.0 logMAR) [16].

Studies on PKMK generally show poorer outcomes than naïve cornea keratitis visual prognosis. For instance, Chatterjee reported logMAR < 1.0 in only 21% of post-treatment PKMK patients [17], while Ong documented mean logMAR of 1.69 at admission, and Ittah-Cohen noted a shift from 1.7 to 0.98 logMAR after treatment [18,19]. Better outcomes have been reported in select cases: Atta observed improvement from 0.98 to 0.44 logMAR in culture- and PCR-negative ulcers, while Cabrera-Aguas reported a VA improvement from 1.7 to 0.98 logMAR in a mixed cohort [4,12,20]. Saeed showed a shift from 0.76 to 0.24 logMAR in contact lens (CL)-wearers [21]. Culture-negative cases were similarly linked to favorable outcomes in Keay’s study [22]. A predictive model for VA outcomes in microbial keratitis identified older age, low initial VA, and corneal transplant history as negative factors, aligning with our findings [15].

Our cohort represents one of the largest post-keratoplasty infectious keratitis groups, directly compared to severe keratitis in virgin corneas. Differences include lower prevalence of CL use and ocular trauma compared to other studies [23,24], high rates of mixed infections (30% vs. ~10% in other large cohorts) [25,26], high glaucoma prevalence, affecting 39% of PKMK and 12% of naïve keratitis patients, likely impairing healing due to long-term antiglaucoma medication use [27].

The longer median hospitalization time for PKMK cases (9 vs. 8 days in the virgin cornea group) reflects general trends [14]. Hospital stay correlated with VA improvement, but no statistical significance was seen in multivariate regression analysis. Seasonal variation showed more severe cases in spring and summer, consistent with prior findings, though final VA was not season-dependent [28,29].

Gram-positive bacteria dominated (52%), consistent with prior research [30]. Notably, our cohort had a high rate of mixed infections (30%), comparable only to Wong’s New Zealand study [26]. Gram-negative bacteria were less common (16%), with PKMK cases showing even lower rates (8% vs. 17% in virgin corneas), reflecting the limited number of CL wearers [21,28,31]. Interestingly, Gram-negative pathogens like *Pseudomonas aeruginosa* have been linked to better final VA outcomes [16,32]. Thus, both high prevalence of mixed infection combined with low incidence of CL wear might contribute to our unfavorable visual results.

There is no universal guideline for microbial keratitis treatment [33,34]. A global survey reported that 20% of ophthalmologists prefer monotherapy (fluoroquinolones), while 78% opt for polytherapy, typically combining aminoglycosides with beta-lactams or vancomycin [35]. Our study found no significant difference in final VA between mono- and polytherapy, except in the PKMK group, where polytherapy led to better outcomes. Nayel demonstrated superior healing rates with gentamicin plus vancomycin compared to moxifloxacin monotherapy or ceftazidime–vancomycin combinations [36]. Given the high prevalence of Gram-positive bacteria and increasing resistance to antibiotics (mainly Methicillin-resistant *Staphylococcus aureus*), our approach of using vancomycin plus aminoglycoside remains optimal [36,37,38,39,40].

In summary, best-corrected visual acuity improved at least one Snellen line in 47% of patients during hospital admission due to corneal ulcer. However, the discrepancy between naïve cornea and transplanted cornea was significant (52% vs. 33%, respectively). Median BCVA improved from 1.9 logMAR to 1.5 logMAR for the entire cohort but the discrepancy between the groups was observed. Younger age, better initial VA and corneal midperiphery location represent positive prognostic factors for the visual outcome. Finally, 67% of patients with corneal ulcer admitted to the hospital were discharged with a legal blindness level of BCVA. Near visual acuity was enhanced in 23% of patients.

The main limitation of this study is the lack of knowledge about baseline visual acuity prior to the infection onset, particularly in transplanted corneas, which might contribute to the initially low VA. Preexisting blindness cannot be excluded. The retrospective nature of the project limited our ability to obtain all corneal culture results (especially from years 2008 to 2015, when solely paper print records from the microbiology department were provided). Furthermore, the majority of our patients were referred to our tertiary center for non-healing ulcers with empirical treatment. Thus, corneal cultures were not obtained due to ongoing extensive antimicrobial therapy. Lastly, as our study design is limited to the hospitalization period, we were unable to assess final VA at the time of ultimately healed ulcers.

Considering the global burden of complications and the occupational and psychological impact on patients’ lives, it is of utmost importance to improve keratitis care. For proper counselling, we need to provide data regarding the level of improvement after hospital admission treatment, including near vision. This study presents one of the largest cohorts of microbial keratitis patients from Central-Eastern Europe, where there is a scarcity of epidemiological studies on corneal ulcers, contributing to the local and ethnic characteristics of several ocular disorders.

## Figures and Tables

**Table 1 jcm-14-00124-t001:** Demographics of the study participants, whole cohort, and two subgroups: Group A (keratitis in naïve cornea) and Group B (post-keratoplasty microbial keratitis).

Variable	Overall (N = 497)	Group A (N = 379)	Group B (N = 118)	*p*-Value
Age (years), Median (Q1–Q3)	61 (43–74)	60 (38.5–72)	64 (49–77)	**0.014** *
Sex female	55.7% (N = 277)	56.5% (N = 214)	53.4% (N = 63)	0.631 **
Eye, right	51.1% (N = 254)	52.2% (N = 198)	47.5% (N = 56)	0.422 **
Days of hospitalization Median (Q1–Q3)	8 (5–12)	8 (5–12.75)	9 (7–12)	0.130 *
Season				
Winter	18.7% (N = 93)	18.7% (N = 71)	18.6% (N = 22)	1 **
Spring	28.8% (N = 143)	28.5% (N = 108)	29.7% (N = 35)	0.898 **
Summer	31.2% (N = 155)	31.7% (N = 120)	29.7% (N = 35)	0.767 **
Autumn	21.3% (N = 106)	21.1% (N = 80)	22% (N = 26)	0.932 **

* U Mann–Whitney test, ** chi-squared test. bold indicates statistically significant *p*-values.

**Table 2 jcm-14-00124-t002:** Visual acuity parameters, divided into distance and near vision, legal blindness defined as best-corrected Visual acuity of logMARε1.0, severe visual impairment as logMAR = 0.5–0.99. Analysis of the whole cohort and two subgroups: Group A (keratitis in naïve cornea) and Group B (post-keratoplasty microbial keratitis). “Improvement” is defined as at least one line of Snellen charts during hospitalization.

Variable	Overall (N = 497)	Group A (N = 379)	Group B (N = 118)	*p*-Value
(I) Distance Visual acuity (BCVA)				
(1) admission				
(a) median (Q1–Q3)	1.9 (0.9–2.3)	1.9 (0.8–2.3)	2.3 (1.5–2.3)	**<0.001** *
(b) logMAR ≥ 1 (%, (n))	74.5% (N = 369)	71% (N = 269)	85.5% (N = 100)	**0.003** **
(c) logMAR = 0.5–0.99 (%, (n))	17.8% (N = 88)	19.6% (N = 74)	12% (N = 14)	**0.004** ***
(d) logMAR < 0.5 (%, (n))	8% (N = 40)	9.5% (N = 36)	3.3% (4)	<0.001 **
(2) discharge				
(a) median (Q1–Q3)	1.5 (0.7–2.3)	1.3 (0.5–1.9)	1.9 (1.3–2.3)	**<0.001** *
(b) logMAR ≥ 1 (%, (n))	66.9% (N = 311)	57% (N = 217)	80% (N = 94)	**0.001**
(c) logMAR = 0.5–0.99 (%, (n))	15.7% (N = 73)	17.5% (N = 62)	9.9% (N = 11)	**<0.001** **
(3) improvement				
(%, (n))	47.4% (N = 221)	51.8% (N = 184)	33.3% (N = 37)	**<0.001** **
(II) Near visual acuity				
(1) admission				
(a) Median (Q1–Q3)	1.5 (0.5–2.36)	1.5 (0.5–2.45)	1.25 (0.5–2.25)	0.622 *
(2) discharge				
Median (Q1–Q3)	0.75 (0.5–1.8)	0.75 (0.5–2)	0.75 (0.62–1.38)	0.707 *
(3) improvement				
% (n)	22.9% (N = 114)	27.4% (N = 104)	8.5% (N = 10)	**<0.001** **
(III) Other				
Legal blindness–change, Median (Q1–Q3)	0 (−0.4–0)	0 (−0.6–0)	0 (−0.3–0)	**0.012** *
Lens status—cataract % (n)	3% (N = 15)	3.7% (N = 14)	0.8% (N = 1)	0.216 ***

* U Mann–Whitney test, ** chi-squared test, *** Fisher test. bold indicates statistically significant *p*-values.

**Table 3 jcm-14-00124-t003:** Clinical characteristic of the cohort. Analysis of the whole cohort and two subgroups: Group A (keratitis in naïve cornea) and Group B (post-keratoplasty microbial keratitis).

Variable	Overall (N = 497)	Group A (N = 379)	Group B (N = 118)	*p*-Value
Contact lens use	14.9% (N = 69)	19.3% (N = 68)	0.9% (N = 1)	<0.001 **
Trauma	11.3% (N = 56)	14% (N = 53)	2.5% (N = 3)	<0.001 **
Active uveitis	13.7% (N = 68)	14.8% (N = 56)	10.2% (N = 12)	0.2635 *
Glaucoma	18.5% (N = 92)	12.1% (N = 46)	39% (N = 46)	**<0.001** *
Part of the cornea affected				
Central	47.5% (N = 235)	46.2% (N = 174)	51.7% (N = 61)	0.344 *
Midperiphery	29.3% (N = 145)	31.3% (N = 118)	22.9% (N = 27)	0.1015 *
Periphery	15.8% (N = 78)	15.1% (N = 57)	17.8% (N = 21)	0.5811 *
Whole cornea	7.5% (N = 37)	7.4% (N = 28)	7.6% (N = 9)	1 *
Ulcer status				
1 lesion	88.9% (N = 441)	89.4% (N = 339)	87.2% (N = 102)	0.6072 *
multifocal	11.1% (N = 55)	10.6% (N = 40)	12.8% (N = 15)	

* Chi-square test, ** Fisher test. Bold indicates statistically significant *p*-values.

**Table 4 jcm-14-00124-t004:** Ancillary testing performed in microbial keratitis diagnosis. Analysis of the whole cohort and two subgroups: Group A (keratitis in naïve cornea) and Group B (post-keratoplasty microbial keratitis).

Variable	Overall (N = 497)	Group A (N = 379)	Group B (N = 118)	*p*-Value
Gram +	52.1% (N = 84)	50% (N = 67)	62.5% (N = 17)	0.3697 *
*Staphylococcus*	67	55	12	
*Streptococcus*	5	4	1	
*Cutibacterium acnes*	4	4		
*Enterococcus faecalis*	2	1	1	
*Bacillus *spp.	4	1	3	
Others	2	2	0	
Gram −	15.8% (N = 23)	17.2% (N = 21)	8.3% (N = 2)	0.3688 **
*Pseudomonas aeruginosa*	15	15		
*Moraxella*	2	2		
*Serratia marcescens *	4	2	2	
Others	2	2		
Fungi	8.2% (N = 12)	9.8% (N = 12)	0% (N = 0)	0.2168 **
*Aspergillus* spp.	4	4		
*Candida* spp.	5	5		
*Fusarium* spp.	3	3		
Mixed infection	24% (N = 35)	23% (N = 28)	29.2% (N = 7)	0.6962 *
Conjunctival swabs (performed)	28.6% (N = 139)	29.4% (N = 109)	26.1% (N = 30)	0.5723 *
Anterior chamber tap (performed)	1.8% (N = 9)	1.8% (N = 7)	1.7% (N = 2)	1 **
CL/suitcase culture (performed)	2% (N = 10)	2.1% (N = 8)	1.7% (N = 2)	1 **
Total organisms growth ^&^	89.0% (N = 145)	84% (N = 120)	21.7% (N = 25)	**0.0414** *
Confocal imaging (performed)	11.1% (N = 55)	12.9% (N = 49)	5.1% (N = 6)	**0.0275** *

^&^ Organism growth was defined as either positive scrapes result or microorganism growth from the anterior chamber tap or conjunctival swab other than natural ocular surface microbiome, * chi-squared test, ** Fisher test. Bold indicates statistically significant *p*-values.

**Table 5 jcm-14-00124-t005:** Clinical and surgical treatment applied in the whole cohort of hospital-admitted corneal ulcer patients, with subgroups analysis: Group A (keratitis in naïve cornea) and Group B (post-keratoplasty microbial keratitis).

Variable	Overall (N = 497)	Group A (N = 379)	Group B (N = 118)	*p*-Value
Antibiotics—monotherapy	30% (N = 149)	31.4% (N = 119)	25.6% (N = 30)	0.2837 *
*Fluoroquinolones*	28.3% (N = 130)	29.7% (N = 102)	24.3% (N = 28)	0.3305 *
*Aminoglycoside*	3.1% (N = 14)	3.8% (N = 13)	0.9% (N = 1)	0.2059 **
*Chloramphenicol*	4.6% (N = 21)	2.6% (N = 9)	10.4% (N = 12)	**0.0013** *
Antibiotics—polytherapy	63.7% (N = 316)	60.9% (N = 231)	72.6% (N = 85)	**0.0285** *
*Gentamicin + Vancomycin*	3.3% (N = 15)	3.8% (N = 13)	1.7% (N = 2)	0.376 **
*Gentamicin + Vancomycin + Fluoroquinolone*	30.5% (N = 140)	30.8% (N = 106)	29.6% (N = 34)	0.8928 *
*Fluoroquinolone + Aminoglicosyde*	24.4% (N = 112)	23.5% (N = 81)	27% (N = 31)	0.5407 *
Others antibacterial therapies	5.9% (N = 27)	5.8% (N = 20)	6.1% (N = 7)	1 *
Antifungal	53.4% (N = 265)	53.2% (N = 201)	54.2% (N = 64)	0.9233 *
Antiviral	19.2% (N = 95)	21.2% (N = 80)	12.7% (N = 15)	0.0556 *
Anti-Amoebal	5.2% (N = 26)	6.6% (N = 25)	0.8% (N = 1)	**0.0153** **
Disinfectants	10.5% (N = 52)	11.9% (N = 45)	6% (N = 7)	0.0968 *
Steroids	63.3% (N = 314)	61.1% (N = 231)	70.3% (N = 83)	0.088 *
Cycloplegics	66.5% (N = 327)	70.6% (N = 266)	53% (N = 61)	**<0.001***
Subtenon Injections	26.5% (N = 129)	29.2% (N = 108)	17.9% (N = 21)	**0.0225** *
*Amniotic membrane*	21.8% (N = 108)	22.3% (N = 84)	20.3% (N = 24)	0.7504 *
*Corneal transplant* (therapeutic)	19.4% (N = 96)	20.6% (N = 78)	15.3% (N = 18)	0.2468 *

* chi-squared test, ** Fisher test. Bold indicates statistically significant *p*-values.

**Table 6 jcm-14-00124-t006:** Univariate regression model (ANCOVA model) for logMAR improvement in the whole cohort of patients (n = 497).

Variable	Estimate	(95% CI)	*p*-Value
Age (years)	0.018	0.012; 0.023	**<0.001**
Days of hospitalization	−0.027	−0.048; −0.005	**0.016**
Distance Visual acuity (BCVA)—admission	−0.483	−0.641; −0.326	**<0.001**
Contact lens use	−0.644	−0.893; −0.395	**<0.001**
Trauma	−0.248	−0.585; 0.090	0.149
Ulcer location: centrum	0.052	−0.204; 0.308	0.689
Ulcer location: midperiphery	−0.322	−0.607; −0.037	**0.027**
Ulcer location: periphery	0.053	−0.313; 0.419	0.776
Ulcer status: multifocal	−0.100	−0.481; 0.280	0.603
Glaucoma	0.684	0.353; 1.015	**<0.001**
Active uveitis	−0.118	−0.428; 0.191	0.451
Corneal transplant (therapeutic)	−0.201	−0.547; 0.145	0.252
Antibiotics treatment: polytherapy	−1.679	−3.077; −0.281	**0.019**
Antibiotics treatment: monotherapy	0.004	−0.376; 0.384	0.983
Antifungal	−0.385	−0.646; −0.125	**0.004**
Culture positive	−0.159	−0.476; 0.157	0.321
Gram + bacteria cultured	0.193	−0.056; 0.443	0.128
Gram − bacteria cultured	−0.106	−0.438; 0.226	0.528
Post-keratoplasty infectious keratitis	0.712	0.388; 1.036	**<0.001**

Bold indicates statistically significant *p*-values.

**Table 7 jcm-14-00124-t007:** Multivariate regression model (ANCOVA model) for logMAR improvement in the whole cohort of patients (n = 497).

Variable	Estimate	(95% CI)	*p*-Value
Intercept	0.029	−0.381; 0.439	0.890
Distance Visual acuity—admission	−0.593	−0.730; −0.455	**<0.001**
Age (years)	0.013	0.008; 0.018	**<0.001**
Sex: Male	0.193	−0.014; 0.400	0.067
Ulcer location: Midperiphery	−0.241	−0.481; −0.001	**0.049**
Glaucoma	0.247	−0.070; 0.563	0.126
Corneal transplant (therapeutic)	−0.278	−0.566; 0.011	0.059
Polytherapy	−1.043	−2.207; 0.121	0.079
Antifungal	−0.187	−0.410; 0.035	0.098
Post-keratoplasty infectious keratitis	0.285	−0.028; 0.598	0.074

Bold indicates statistically significant *p*-values.

## Data Availability

The data presented in this study are available on request from the corresponding author.

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
