# Peer review of "Severe Microbial Keratitis in Virgin and Transplanted Cornea—Probability of Visual Acuity Improvement"

_jcm, 2024, doi:10.3390/jcm14010124_

Round 1
Reviewer 1 Report
Comments and Suggestions for Authors
line 23 the BCVA did not "decrease" (it is true that logMAR is smaller as VA is better). Please find another phrasing that is less confusing.
Also, please avoid the repetition "and decreased"..."and improved".
There is inconsistency between the abstract "Legal blindness status changed for 52/379 (14%) patients in group A 27 and 6/118 (5%) patients in group B during hospital admission (p<0,001)." and Table 2: in group A 71.2% at admission and 61.3% at discharge, in group B 85.5% at admission, 84.7% at discharge. While in the Discussion section: "During
220
hospital admission, legal blindness status changed in 18% of the virgin cornea group and only 6% of the PKMK group." Which is correct??
line 30 "other than midperiphery location ... independent negative risk factors for improvement" - does this mean that peripheral ulcers have a worse prognosis than mid-peripheral ulcers? In fact, you should write what ANCOVA has shown, i.e. that mid-peripheral ulcers have a better prognosis (probably there were too few peripheral ulcers)
line 97-98 "Keratoplasty, amniotic membrane transplantation and evisceration rate was excluded from the notes." - but you have communicated the rates of amniotic transplantations and eviscerations
line 259 "Longer hospitalisation correlated with more favourable clinical outcomes in our cohort." - this affirmation is not supported by ANCOVA results
Author Response
Dear Reviewer,
We are grateful for your detailed comments on our manuscript. Thank you for your time and the effort it took to review our manuscript. We are more than grateful for the opportunity to improve our paper with the modifications you suggested.We have restructured the paper to address these comments as follows:
Reviewers` comments:
Comments 1: line 23 the BCVA did not "decrease" (it is true that logMAR is smaller as VA is better). Please find another phrasing that is less confusing.
Response 1: Thank you for your comment. We corrected the sentence as well as the whole abstract to make the be less confusing. Please see below:
“The median BCVA at presentation for the entire cohort was 1.9 logMAR, advancing to 1.5 logMAR at discharge (p<0.001). At least one line improvement in BCVA was observed in 47% of patients (52% of naïve cornea and 33% of transplanted cornea patients).” (line 51-54)
Comments 2: Also, please avoid the repetition "and decreased"..."and improved".
Response 2: Thank you for your comment. We have now amended the manuscript throughout to replace repetitions with the synonyms of “decreased” (dropped, reduced, lessened) as well as “improved” visual acuity (advancing to, enhanced, got better, picked up). We have also rearranged the sentences with new grammatical structures to replace repetitions.
Comments 3: There is inconsistency between the abstract "Legal blindness status changed for 52/379 (14%) patients in group A 27 and 6/118 (5%) patients in group B during hospital admission (p<0,001)." and Table 2: in group A 71.2% at admission and 61.3% at discharge, in group B 85.5% at admission, 84.7% at discharge. While in the Discussion section: "During hospital admission, legal blindness status changed in 18% of the virgin cornea group and only 6% of the PKMK group." Which is correct??
Response 3: Thank you for your comment. We have double-checked the results and admit to having mistakenly calculated percent of legal blindness patients in group B at discharge. We are very sorry for that mistake. Legal blindness level of vision was observed in 67% of patients in the group A and 80% of patients in group B at discharge. We recalculated the results and modified the discussion section accordingly. (Table 2- please see below). Of note, Additionally, the Table 2 was reorganized to improve clarity of the results.
Variable |
Overall (N=497) |
group A (N=379) |
group B (N=118) |
p-value |
I) Distance Visual acuity (BCVA) |
||||
1) admission a) median (Q1-Q3) b) logMAR≥1 (%, (n)) c) logMAR= 0.5 – 0.99 (%, (n)) d) logMAR <0.5 (%, (n) |
1.9 (0.9 - 2.3) |
1.9 (0.8 - 2.3) |
2.3 (1.5 - 2.3) |
<0.001* |
74.5% (N=369) |
71% (N=269) |
85.5% (N=100) |
0.003** 0.004***
<0.001** |
|
17.8% (N=88)
8% (N=40) |
19.6% (N=74)
9.5% (N=36) |
12% (N=14)
3.3 % (4) |
||
2) discharge a) median (Q1-Q3) b) logMAR≥1 (%, (n) c) logMAR= 0.5 – 0.99 (%, (n) |
1.5 (0.7 – 2.3) |
1.3 (0.5 – 1.9) |
1.9 (1.3 – 2.3) |
<0.001* |
66.9% (N=311) |
57% (N=217) |
80% (N=94) |
0.001 <0.001** |
|
15.7% (N=73) |
17.5% (N=62) |
9.9% (N=11) |
||
3) improvement (%, (n) |
47.4% (N=221) |
51.8% (N=184) |
33.3% (N=37) |
<0.001** |
II) Near visual acuity |
||||
1) admission a) Median (Q1-Q3) |
1.5 (0.5 - 2.36) |
1.5 (0.5 - 2.45) |
1.25 (0.5 - 2.25) |
0.622* |
2) discharge Median (Q1-Q3) |
0.75 (0.5 - 1.8)
|
0.75 (0.5 - 2) |
0.75 (0.62 - 1.38) |
0.707* |
3) improvement % (n) |
22.9% (N=114) |
27.4% (N=104) |
8.5% (N=10) |
<0.001** |
III) Other |
||||
Legal blindness – change, Median (Q1-Q3) |
0 (-0.4 - 0) |
0 (-0.6 - 0) |
0 (-0.3 - 0) |
0.012* |
Lens status – cataract % (n) |
3% (N=15) |
3.7% (N=14) |
0.8% (N=1) |
0.216*** |
Comments 4: line 30 "other than midperiphery location ... independent negative risk factors for improvement" - does this mean that peripheral ulcers have a worse prognosis than mid-peripheral ulcers? In fact, you should write what ANCOVA has shown, i.e. that mid-peripheral ulcers have a better prognosis (probably there were too few peripheral ulcers)
Response 4: Thank you for your comment and we are sorry for this ambiguity. We corrected the line according to your suggestions.
Comments 5: line 97-98 "Keratoplasty, amniotic membrane transplantation and evisceration rate was excluded from the notes." - but you have communicated the rates of amniotic transplantations and eviscerations
Response 5: Thank you for your comment. We corrected the sentence accordingly. We are very sorry for this semantic mistake (our aim was to use the word “extracted” instead of “excluded”).
Comments 6: line 259 "Longer hospitalisation correlated with more favourable clinical outcomes in our cohort." - this affirmation is not supported by ANCOVA results
Response 6: Thank you for your comment. We based this sentence on the univariate regression model, but the multivariate regression model did not support the finding. We corrected the abstract, results and discussion section accordingly.
Additionally, we re-organised the results section to improve the flow and underline crucial findings.
We again would like to thank you for your valuable feedback and hope that you accept our incorporation of the comments into our amended manuscript.
Sincerely,
Joanna Przybek-Skrzypecka, MD, PhD
Corresponding author

Reviewer 2 Report
Comments and Suggestions for Authors
I appreciate the opportunity to review this interesting article on assessing visual acuity in patients with microbial keratitis. Though the authors tried to present the results and facts up to certain level but to me (may be other scientific readers) it would be difficult to understand what the authors trying to convey with their findings. My comments are below:
Is it 497 or 517 patients? Need to be verified.
Line 50: "However, ophthalmology is unique" I couldn't understand this line.
Line 108: Confocal microscopy analysis mentioned but there is no representative image of that.
Line 126-128: Repetition of sentences. Correct this
Table 1 seems to be very unorganized
Results 3.2 Visual acuity. Table not referenced within the result. Also, I failed to corelate results with the table provided. Same for results 3.3.
The authors mentioned group A and group B in their results, but it is difficult to get this clarification from the tables provided. I would suggest authors to work on the tables or redraw the tables.
Reference mentioned at most places after full stop. Check this and correct.
My major suggestion to the authors would be to work on results sections along with discussion and tables more to make the manuscript comprehensible for the reader.
Comments on the Quality of English LanguageThe English in the present manuscript is not of publication quality and require some improvement and proper proof-reading.
Author Response
Dear Reviewer,
We are grateful for Your detailed comments on our manuscript. Please find the answers below and in the revised manuscript (all the recommended modifications marked in yellow).
Reviewers` comments:
General comment:
I appreciate the opportunity to review this interesting article on assessing visual acuity in patients with microbial keratitis. Though the authors tried to present the results and facts up to certain level but to me (may be other scientific readers) it would be difficult to understand what the authors trying to convey with their findings. My comments are below:
Re: Thank you for your time and effort to review our manuscript. We are more than grateful for the opportunity to improve our paper with modifications you suggested.
Specific comments:
Comments 1: Is it 497 or 517 patients? Need to be verified.
Response 1: Thank you for pointing out this discrepancy. After careful consideration, we decided to use the final analyzed cohort of 497 patients in the abstract and methods section to ensure consistency and clarity. The primary number of 517 patients included cases initially classified as microbial keratitis, but after a thorough review, 20 cases were excluded as they were determined to be non-infectious, immunological ulcers.
This unification reflects the accurate number of cases analyzed for this study, providing a more precise representation of the data and ensuring the results focus solely on infectious keratitis. Below, you will find the updated text from the abstract and methods sections, which has been modified to reflect this decision:
Abstract:
“A retrospective analysis was conducted on 497 patients with unilateral corneal ulcers treated at a tertiary referral center from 2008 to 2023” (line 41-42).
Methods:
“A retrospective analysis was conducted on 517 adults diagnosed with severe microbial keratitis, identified using the International Classification of Diseases (ICD-10) code H16.1, and admitted to the tertiary eye center at the Department of Ophthalmology, Medical University of Warsaw, Poland, between 2008 and 2023. After excluding 20 cases initially coded as microbial keratitis but later diagnosed as non-infectious or immunological ulcers, the final analysis included 497 cases.”
(line 104-109)
We hope this explanation clarifies our decision to standardize the patient cohort and accurately represent the study’s scope. Thank you again for bringing this to our attention.
Comments 2: Line 50: "However, ophthalmology is unique" I couldn't understand this line.
Response 2: Thank you for that comment. We have expanded and reworded the whole paragraph to clarify our idea. Please see it below:
“Given the limited number of inpatient beds in modern ophthalmology departments as well as the high cost of hospitalization, it is crucial to carefully assess and prioritize conditions that would benefit most from admission [5,6]. Although direct evidence in ophthalmology is yet to be established, other direct cost-benefit relationships have been shown for other conditions, namely myocardial infarction and severe pneumonia [7-10].” (lines: 84-89)
Comments 3: Line 108: Confocal microscopy analysis mentioned but there is no representative image of that.
Response 3: Thank you for your valuable comment. We appreciate your suggestion regarding the inclusion of confocal microscopy images. While we initially considered including representative images, we ultimately decided that doing so would exceed the scope of this article, which already addresses a wide range of issues. However, we recognize the importance of such visual data and are planning to incorporate confocal microscopy findings in a future publication focused specifically on imaging aspects of keratitis (one of aforementioned projects has been currently reviewed in Scientific Reports journal). We hope this explanation clarifies our decision.
Comments 4: Line 126-128: Repetition of sentences. Correct this
Response 4: We fully agree with your comment, and we are very sorry to waste your time. We corrected this accordingly into one sentence: “Multiple ANCOVA regression models were used to examine the relationship between the dependent variable and explanatory variables for final visual acuity of legal blindness.”
Comments 5: Table 1 seems to be very unorganized
Response 5: We also fully agree with your comment, and we are very sorry to waste your time. We decided to reorganise Table 1 with several amendments:
- The first and second columns were unified with the parameter of variable placed underneath it
- We shortened the headings of the groups A and B
- The column presenting statistical test applied to the variable was removed and replaced it into superscript’ s stars system explained below the Table
- We decided to remove the effect size column
- The seasons’ lines were unified and combined to the core of the Table 1.
Table 1. Demographics of the study participants, whole cohort and two subgroups: group A- keratitis in naïve cornea and group B- post-keratoplasty microbial keratitis.
Variable |
Overall (N=497) |
group A (N=379) |
group B (N=118) |
p-value |
Age (years), Median (Q1-Q3) |
61 (43 - 74) |
60 (38.5 - 72) |
64 (49 - 77) |
0.014 * |
Sex female |
55.7% (N=277) |
56.5% (N=214) |
53.4% (N=63) |
0.631 ** |
Eye, right |
51.1% (N=254) |
52.2% (N=198) |
47.5% (N=56) |
0.422 ** |
Days of hospitalization Median (Q1-Q3) |
8 (5 - 12) |
8 (5 - 12.75) |
9 (7 - 12) |
0.130 * |
Season |
|
|||
Winter |
18.7% (N=93) |
18.7% (N=71) |
18.6% (N=22) |
1** |
Spring |
28.8% (N=143) |
28.5% (N=108) |
29.7% (N=35) |
0.898** |
Summer |
31.2% (N=155) |
31.7% (N=120) |
29.7% (N=35) |
0.767** |
Autumn |
21.3% (N=106) |
21.1% (N=80) |
22% (N=26) |
0.932** |
*U Mann-Whitney test, **chi-squared test
bold indicates statistically significant p-values
Comments 6: Results 3.2 Visual acuity. Table not referenced within the result. Also, I failed to corelate results with the table provided. Same for results 3.3.
Response 6: Thank you for that comment and we are very sorry for this basic mistake. We have referred the Table in the text as well as we double-checked the database and statistical analysis finding a mistake. We addressed all the issues mentioned in the re-organised Table 2- please see it below.
Table 2. Visual acuity parameters, divided into distance and near vision, legal blindness defined as best-corrected Visual acuity of logMAR≥1.0, severe visual impairment as logMAR= 0.5-0.99. Analysis of the whole cohort and two subgroups: group A- keratitis in naïve cornea and group B- post-keratoplasty microbial keratitis. “Improvement” defined as at least one line of Snellen charts during hospitalization.
Variable |
Overall (N=497) |
group A (N=379) |
group B (N=118) |
p-value |
I) Distance Visual acuity (BCVA) |
||||
1) admission a) median (Q1-Q3) b) logMAR≥1 (%, (n)) c) logMAR= 0.5 – 0.99 (%, (n)) d) logMAR <0.5 (%, (n) |
1.9 (0.9 - 2.3) |
1.9 (0.8 - 2.3) |
2.3 (1.5 - 2.3) |
<0.001* |
74.5% (N=369) |
71% (N=269) |
85.5% (N=100) |
0.003** 0.004***
<0.001** |
|
17.8% (N=88)
8% (N=40) |
19.6% (N=74)
9.5% (N=36) |
12% (N=14)
3.3 % (4) |
||
2) discharge a) median (Q1-Q3) b) logMAR≥1 (%, (n) c) logMAR= 0.5 – 0.99 (%, (n) |
1.5 (0.7 – 2.3) |
1.3 (0.5 – 1.9) |
1.9 (1.3 – 2.3) |
<0.001* |
66.9% (N=311) |
57% (N=217) |
80% (N=94) |
0.001 <0.001** |
|
15.7% (N=73) |
17.5% (N=62) |
9.9% (N=11) |
||
3) improvement (%, (n) |
47.4% (N=221) |
51.8% (N=184) |
33.3% (N=37) |
<0.001** |
II) Near visual acuity |
||||
1) admission a) Median (Q1-Q3) |
1.5 (0.5 - 2.36) |
1.5 (0.5 - 2.45) |
1.25 (0.5 - 2.25) |
0.622* |
2) discharge Median (Q1-Q3) |
0.75 (0.5 - 1.8)
|
0.75 (0.5 - 2) |
0.75 (0.62 - 1.38) |
0.707* |
3) improvement % (n) |
22.9% (N=114) |
27.4% (N=104) |
8.5% (N=10) |
<0.001** |
III) Other |
||||
Legal blindness – change, Median (Q1-Q3) |
0 (-0.4 - 0) |
0 (-0.6 - 0) |
0 (-0.3 - 0) |
0.012* |
Lens status – cataract % (n) |
3% (N=15) |
3.7% (N=14) |
0.8% (N=1) |
0.216*** |
*U Mann-Whitney test, **chi-squared test, *** Fisher test
bold indicates statistically significant p-values
Comments 7: The authors mentioned group A and group B in their results, but it is difficult to get this clarification from the tables provided. I would suggest authors to work on the tables or redraw the tables.
Response 7: Thank you for that comment, we have worked extensively on the tables, redrawing and reorganizing them. We introduced this division into column headings of each Table and explain the meaning of the “group A” and “group B” at the top of the Tables. Please find it in the Table 3 below (presented as an example).
Table 3. Clinical characteristic of the cohort. Analysis of the whole cohort and two subgroups: group A- keratitis in naïve cornea and group B- post-keratoplasty microbial keratitis.
Variable |
Overall (N=497) |
group A (N=379) |
group B (N=118) |
p-value |
Contact lens use |
14.9% (N=69) |
19.3% (N=68) |
0.9% (N=1) |
<0.001** |
Trauma |
11.3% (N=56) |
14% (N=53) |
2.5% (N=3) |
<0.001** |
Active uveitis |
13.7% (N=68) |
14.8% (N=56) |
10.2% (N=12) |
0.2635* |
Glaucoma |
18.5% (N=92) |
12.1% (N=46) |
39% (N=46) |
<0.001* |
Part of the cornea affected Central Midperiphery Periphery Whole cornea
|
47.5% (N=235) |
46.2% (N=174) |
51.7% (N=61) |
0.344* |
29.3% (N=145) |
31.3% (N=118) |
22.9% (N=27) |
0.1015* |
|
15.8% (N=78) |
15.1% (N=57) |
17.8% (N=21) |
0.5811* |
|
7.5% (N=37) |
7.4% (N=28) |
7.6% (N=9) |
1* |
|
Ulcer status 1 lesion multifocal |
88.9% (N=441) |
89.4% (N=339) |
87.2% (N=102) |
0.6072* |
11.1% (N=55) |
10.6% (N=40) |
12.8% (N=15) |
Comments 8: Reference mentioned at most places after full stop. Check this and correct.
Response 8: We fully agree with your comments. All the references were checked and cited accordingly to the journal guidelines (placed inside the punctuation).
Comments 9: My major suggestion to the authors would be to work on results sections along with discussion and tables more to make the manuscript comprehensible for the reader.
Response 9: Thank you for your valuable comment. We appreciate your suggestion and have been working extensively on the results along with discussion sections to clarify our ideas. The abstract was also rewritten to focus more on our primary goal (particularly the results and conclusion sections). Furthermore, we concentrated on our main message (visual acuity amendments) in the results and discussion section. Significant parts of the Table 2, 3 and 4 were removed not to distract the reader from the main message of the article. Each table has been modified to improve legibility and the sixth table was entirely changed for univariate analysis of the factors influencing final BCVA. Finally, discussion was enriched with the summary paragraph to underline our main message. We hope this explanation clarifies our decision and meet your expectations. Please find the amended “results” and “discussion” section below.
“Results
Demographics
The analysis included 497 patients admitted to a tertiary eye hospital with a clinical diagnosis of infectious corneal ulcer. Of these, 55.7% were female. The median age was 61. The study included 379 patients with keratitis in naïve corneas (Group A) and 118 post-keratoplasty microbial keratitis patients (Group B). There were no significant demographic differences between the two groups. Seasonal variations were noted, with a higher incidence of keratitis cases occurring in the summer and spring. Table 1 presents the demographic characteristics in detail.
Table 1. Demographics of the study participants, whole cohort and two subgroups: group A- keratitis in naïve cornea and group B- post-keratoplasty microbial keratitis.
Variable |
Overall (N=497) |
group A (N=379) |
group B (N=118) |
p-value |
Age (years), Median (Q1-Q3) |
61 (43 - 74) |
60 (38.5 - 72) |
64 (49 - 77) |
0.014 * |
Sex female |
55.7% (N=277) |
56.5% (N=214) |
53.4% (N=63) |
0.631 ** |
Eye, right |
51.1% (N=254) |
52.2% (N=198) |
47.5% (N=56) |
0.422 ** |
Days of hospitalization Median (Q1-Q3) |
8 (5 - 12) |
8 (5 - 12.75) |
9 (7 - 12) |
0.130 * |
Season |
|
|||
Winter |
18.7% (N=93) |
18.7% (N=71) |
18.6% (N=22) |
1** |
Spring |
28.8% (N=143) |
28.5% (N=108) |
29.7% (N=35) |
0.898** |
Summer |
31.2% (N=155) |
31.7% (N=120) |
29.7% (N=35) |
0.767** |
Autumn |
21.3% (N=106) |
21.1% (N=80) |
22% (N=26) |
0.932** |
*U Mann-Whitney test, **chi-squared test
bold indicates statistically significant p-values
Visual acuity
The median visual acuity on admission was 1.9 logMAR (Q1:Q3= 0.9- 2.3) for the entire cohort, with a significant difference between the groups (Group A: 1.9 logMAR, Group B: 2.3 logMAR, p<0.001). An improvement of at least one Snellen line was observed in 52% of patients in Group A and 33% in Group B (p<0.001). The influence of cataracts was excluded as no significant difference in lens status was noted (p= 0.21). Legal blindness status altered for 11.7% of the cohort from admission to discharge (71% vs 57% in Group A and 86% vs 80% in Group B), with significant improvement only in the naïve keratitis group. Of note, median BCVA among legal blindness patients was 2.3 logMAR for the whole cohort (Q1-Q3 1.6-2.3), group A (Q1-Q3 1.6-2.3), group B (Q1-Q3 1.9-2.3) (p=0,19) and shifted to median 1.9 logMAR for the entire group. Any level of near vision was observed in 134 persons (27% of 497) on admission and 173 (35% of 497) at discharge. Near vision improved in 27% of patients in Group A and 9% in Group B (p<0.001). Table 2 presents details of visual acuity parameters and their shift in the studied period of time. Furthermore, additional analysis based on visual acuity levels: (a) logMAR³1, b) logMAR= 0.5-0.99, c) logMAR<0.5) was performed. Of note, there were statistically significant difference for age, CL use, corneal part affected (central, midperiphery, peripheral), mono/multifocal involvement, active uveitis at presentation, glaucoma history, therapeutic corneal transplant and antifungal treatment. Median age for a) logMAR³1 was 67 years, b) logMAR= 0.5-0.99 – 51 years and c) logMAR<0.5 =35 years, p<0.001). CL use prevalence increases with initial BCVA (9% in logMAR³1 group, 25% in logMAR= 0.5-0.99 and 33% in logMAR<0.5 group.
Table 2. Visual acuity parameters, divided into distance and near vision, legal blindness defined as best-corrected Visual acuity of logMAR≥1.0, severe visual impairment as logMAR= 0.5-0.99. Analysis of the whole cohort and two subgroups: group A- keratitis in naïve cornea and group B- post-keratoplasty microbial keratitis. “Improvement” defined as at least one line of Snellen charts during hospitalization.
Variable |
Overall (N=497) |
group A (N=379) |
group B (N=118) |
p-value |
I) Distance Visual acuity (BCVA) |
||||
1) admission a) median (Q1-Q3) b) logMAR≥1 (%, (n)) c) logMAR= 0.5 – 0.99 (%, (n)) d) logMAR <0.5 (%, (n) |
1.9 (0.9 - 2.3) |
1.9 (0.8 - 2.3) |
2.3 (1.5 - 2.3) |
<0.001* |
74.5% (N=369) |
71% (N=269) |
85.5% (N=100) |
0.003** 0.004***
<0.001** |
|
17.8% (N=88)
8% (N=40) |
19.6% (N=74)
9.5% (N=36) |
12% (N=14)
3.3 % (4) |
||
2) discharge a) median (Q1-Q3) b) logMAR≥1 (%, (n) c) logMAR= 0.5 – 0.99 (%, (n) |
1.5 (0.7 – 2.3) |
1.3 (0.5 – 1.9) |
1.9 (1.3 – 2.3) |
<0.001* |
66.9% (N=311) |
57% (N=217) |
80% (N=94) |
0.001 <0.001** |
|
15.7% (N=73) |
17.5% (N=62) |
9.9% (N=11) |
||
3) improvement (%, (n) |
47.4% (N=221) |
51.8% (N=184) |
33.3% (N=37) |
<0.001** |
II) Near visual acuity |
||||
1) admission a) Median (Q1-Q3) |
1.5 (0.5 - 2.36) |
1.5 (0.5 - 2.45) |
1.25 (0.5 - 2.25) |
0.622* |
2) discharge Median (Q1-Q3) |
0.75 (0.5 - 1.8)
|
0.75 (0.5 - 2) |
0.75 (0.62 - 1.38) |
0.707* |
3) improvement % (n) |
22.9% (N=114) |
27.4% (N=104) |
8.5% (N=10) |
<0.001** |
III) Other |
||||
Legal blindness – change, Median (Q1-Q3) |
0 (-0.4 - 0) |
0 (-0.6 - 0) |
0 (-0.3 - 0) |
0.012* |
Lens status – cataract % (n) |
3% (N=15) |
3.7% (N=14) |
0.8% (N=1) |
0.216*** |
*U Mann-Whitney test, **chi-squared test, *** Fisher test
bold indicates statistically significant p-values
Risk factors and clinical characteristics
Contact lens wearers constituted 15% of the cohort, with a significant difference between Group A and Group B (19% vs 1%, p<0.001). Traumatic events occurred in 56 patients, with 14% in Group A and 3% in Group B (p<0.001). The median time from symptom onset to hospital admission was 9 days for the entire cohort, with Group B patients tending to seek specialist care sooner (a median of 7 days, Q1:Q3= 3-14 days) than Group A patients (a median of 9 days, Q1:Q3= 3-30 days). More patients in Group B received prior treatment from local ophthalmologists (57% in Group B vs 52% in Group A). There were no significant differences in uveitis status between the groups (15% vs 10%), but a significant difference was noted for glaucoma status (12% in Group A vs 39% in Group B, p<0.001). Table 3 depicts further clinical characteristics including the percentage of patients with a certain number and specific locations of infiltrates showing no statistically significant difference between the subgroups.
Table 3. Clinical characteristic of the cohort. Analysis of the whole cohort and two subgroups: group A- keratitis in naïve cornea and group B- post-keratoplasty microbial keratitis.
Variable |
Overall (N=497) |
group A (N=379) |
group B (N=118) |
p-value |
Contact lens use |
14.9% (N=69) |
19.3% (N=68) |
0.9% (N=1) |
<0.001** |
Trauma |
11.3% (N=56) |
14% (N=53) |
2.5% (N=3) |
<0.001** |
Active uveitis |
13.7% (N=68) |
14.8% (N=56) |
10.2% (N=12) |
0.2635* |
Glaucoma |
18.5% (N=92) |
12.1% (N=46) |
39% (N=46) |
<0.001* |
Part of the cornea affected Central Midperiphery Periphery Whole cornea
|
47.5% (N=235) |
46.2% (N=174) |
51.7% (N=61) |
0.344* |
29.3% (N=145) |
31.3% (N=118) |
22.9% (N=27) |
0.1015* |
|
15.8% (N=78) |
15.1% (N=57) |
17.8% (N=21) |
0.5811* |
|
7.5% (N=37) |
7.4% (N=28) |
7.6% (N=9) |
1* |
|
Ulcer status 1 lesion multifocal |
88.9% (N=441) |
89.4% (N=339) |
87.2% (N=102) |
0.6072* |
11.1% (N=55) |
10.6% (N=40) |
12.8% (N=15) |
*Chi-square test, **Fisher test
bold indicates statistically significant p-values
Ancillary tests
Corneal scrapes were performed in 163 patients. Positive corneal culture occurs in 103 patients from the group B (72%) and 11 patients in the group B (55%) (p<0.001, effect size 0.118). In group A we obtained 79 mono-organism growth and 24 multi-bacterial/fungal growth. Any microorganism growth (derived either from scrapes, anterior chamber tap, or swabs other than natural ocular surface microbiome) was detected in 84% of the group A. Microbial growth was proven in 32% for the whole of group A and 22% in group B (p=0,041). Gram-positive bacteria dominated in both subgroups (50% in group A and 63% in group B, p=0.37) followed by mixed infections (group A: 23%, group B: 29%, p=0.70). Almost 10% difference in occurrence was noted for both Gram-negative and fungal infection. Notably, for the whole cohort the percentage of Gram-positive cultures was higher for the elderly group (minimum 65 years old) than in the younger (<65 years old) population (62% vs 46%, respectively, p=0.09). Confocal imaging was obtained more often in the naïve cornea group than PKMK group (13% vs 5%, p=0.028). Table 4 depicts diagnostic details of the subgroups.
Table 4. Ancillary testing performed in microbial keratitis diagnosis. Analysis of the whole cohort and two subgroups: group A- keratitis in naïve cornea and group B- post-keratoplasty microbial keratitis.
Variable |
Overall (N=497) |
group A (N=379) |
group B (N=118) |
p-value |
Gram + |
52.1% (N=84) |
50% (N=67) |
62.5% (N=17) |
0.3697* |
Staphylococcus |
67 |
55 |
12 |
|
Streptococcus |
5 |
4 |
1 |
|
Cutibacterium acnes |
4 |
4 |
|
|
Enterococcus faecalis |
2 |
1 |
1 |
|
Bacillus spp. |
4 |
1 |
3 |
|
Others |
2
|
2
|
0 |
|
Gram – |
15.8% (N=23) |
17.2% (N=21) |
8.3% (N=2) |
0.3688** |
Pseudomonas aeruginosa |
15 |
15 |
|
|
Moraxella |
2 |
2 |
|
|
Serratia marcescens |
4 |
2 |
2 |
|
Others |
2 |
2 |
|
|
Fungi |
8.2% (N=12) |
9.8% (N=12) |
0% (N=0) |
0.2168** |
Aspergillus spp. |
4 |
4 |
|
|
Candida spp. |
5 |
5 |
|
|
Fusarium spp. |
3 |
3 |
|
|
Mixed infection |
24% (N=35) |
23% (N=28) |
29.2% (N=7) |
0.6962* |
Conjunctival swabs (performed) |
28.6% (N=139) |
29.4% (N=109) |
26.1% (N=30) |
0.5723* |
Anterior chamber tap (performed) |
1.8% (N=9) |
1.8% (N=7) |
1.7% (N=2) |
1** |
CL/suitcase culture (performed) |
2% (N=10) |
2.1% (N=8) |
1.7% (N=2) |
1** |
Total organisms growth& |
89,0% (N=145) |
84% (N=120) |
21.7% (N=25) |
0.0414* |
Confocal imaging (performed) |
11.1% (N=55) |
12.9% (N=49) |
5.1% (N=6) |
0.0275* |
&Organism growth was defined as either: positive scrapes result or microorganism growth from the anterior chamber tap or conjunctival swab other than natural ocular surface microbiome
*chi-squared test, ** Fisher test
bold indicates statistically significant p-values
Treatment
Empirical antibiotic therapy was administered in 94% of patients, with polytherapy being more common than monotherapy (61% in Group A vs 73% in Group B, p=0.0285). Polytherapy was associated with a higher probability of visual acuity improvement in the univariate regression model (p=0.019). Fungal empirical treatment was introduced in 53% of patients with no statistically significant difference between the groups. However, univariate regression model proved better visual outcomes in the subgroup of patients with ani-fungal treatment started at the admission. Empirical treatment for Acanthamoeba keratitis was more commonly introduced in Group A (25 vs 1 case in Group B, p=0.015). Cycloplegics were applied more frequently in Group A (71% vs 53%, p<0.001). Surgical intervention was performed in over 40% of the cohort (therapeutic corneal transplant in 19% of patients and amniotic membrane transplantation in 22 %), with no significant difference between the groups. Two patients in the whole cohort required evisceration, both in the naïve cornea group. Table 5 demonstrates details of varied modes of treatment in our study.
Table 5. Clinical and surgical treatment applied in the whole cohort of hospital-admitted corneal ulcer patients, with subgroups analysis (group A- keratitis in naïve cornea and group B- post-keratoplasty microbial keratitis).
Variable |
Overall (N=497) |
group A (N=379) |
group B (N=118) |
p-value |
Antibiotics - monotherapy Fluoroquinolones Aminoglycoside Chloramphenicol |
30% (N=149) |
31.4% (N=119) |
25.6% (N=30) |
0.2837* |
28.3% (N=130) |
29.7% (N=102) |
24.3% (N=28) |
0.3305* |
|
3.1% (N=14) |
3.8% (N=13) |
0.9% (N=1) |
0.2059** |
|
4.6% (N=21) |
2.6% (N=9) |
10.4% (N=12) |
0.0013* |
|
Antibiotics- polytherapy Gentamicin + Vancomycin Gentamicin + Vancomycin + Fluoroquinolone Fluoroquinolone + Aminoglicosyde |
63.7% (N=316) |
60.9% (N=231) |
72.6% (N=85) |
0.0285* |
3.3% (N=15) |
3.8% (N=13) |
1.7% (N=2) |
0.376** |
|
30.5% (N=140) |
30.8% (N=106) |
29.6% (N=34) |
0.8928* |
|
24.4% (N=112) |
23.5% (N=81) |
27% (N=31) |
0.5407* |
|
Others antibacterial therapies |
5.9% (N=27) |
5.8% (N=20) |
6.1% (N=7) |
1* |
Antifungal |
53.4% (N=265) |
53.2% (N=201) |
54.2% (N=64) |
0.9233* |
Antiviral |
19.2% (N=95) |
21.2% (N=80) |
12.7% (N=15) |
0.0556* |
Anti-Amoebal |
5.2% (N=26) |
6.6% (N=25) |
0.8% (N=1) |
0.0153** |
Disinfectants |
10.5% (N=52) |
11.9% (N=45) |
6% (N=7) |
0.0968* |
Steroids |
63.3% (N=314) |
61.1% (N=231) |
70.3% (N=83) |
0.088* |
Cycloplegics |
66.5% (N=327) |
70.6% (N=266) |
53% (N=61) |
<0.001* |
Subtenon Injections |
26.5% (N=129) |
29.2% (N=108) |
17.9% (N=21) |
0.0225* |
Amniotic membrane |
21.8% (N=108) |
22.3% (N=84) |
20.3% (N=24) |
0.7504* |
Corneal transplant (therapeutic) |
19.4% (N=96) |
20.6% (N=78) |
15.3% (N=18) |
0.2468* |
*chi-squared test, ** Fisher test
bold indicates statistically significant p-values
Table 6 presents the results of univariate regression model for visual acuity improvement (defined as at least one line on Snellen charts) for the whole cohort. It occurred that: longer hospitalization time, better BCVA at presentation, CL use, midperipheral location, antibacterial polytherapy treatment and antifungal treatment were statistically significant for better visual outcomes. On the contrary, older age, history of glaucoma and post-keratoplasty corneal ulcer represent negative predictive factors.
Table 6. Univariate regression model (ANCOVA model) for logMAR improvement in the whole cohort of patients (n=497).
Variable |
Estimate |
(95% CI) |
p-value |
Age (years) |
0.018 |
0.012; 0.023 |
<0.001 |
Days of hospitalization |
-0.027 |
-0.048; -0.005 |
0.016 |
Distance Visual acuity (BCVA) - admission |
-0.483 |
-0.641; -0.326 |
<0.001 |
Contact lens use |
-0.644 |
-0.893; -0.395 |
<0.001 |
Trauma |
-0.248 |
-0.585; 0.090 |
0.149 |
Ulcer location: centrum |
0.052 |
-0.204; 0.308 |
0.689 |
Ulcer location: midperiphery* |
-0.322 |
-0.607; -0.037 |
0.027 |
Ulcer location: periphery |
0.053 |
-0.313; 0.419 |
0.776 |
Ulcer status: multifocal |
-0.100 |
-0.481; 0.280 |
0.603 |
Glaucoma |
0.684 |
0.353; 1.015 |
<0.001 |
Active uveitis |
-0.118 |
-0.428; 0.191 |
0.451 |
Corneal transplant (therapeutic) |
-0.201 |
-0.547; 0.145 |
0.252 |
Antibiotics treatment: polytherapy |
-1.679 |
-3.077; -0.281 |
0.019 |
Antibiotics treatment: monotherapy |
0.004 |
-0.376; 0.384 |
0.983 |
Antifungal |
-0.385 |
-0.646; -0.125 |
0.004 |
Culture positive |
-0.159 |
-0.476; 0.157 |
0.321 |
Gram + bacteria cultured |
0.193 |
-0.056; 0.443 |
0.128 |
Gram - bacteria cultured |
-0.106 |
-0.438; 0.226 |
0.528 |
Post-keratoplasty infectious keratitis |
0.712 |
0.388; 1.036 |
<0.001 |
bold indicates statistically significant p-values
Furthermore, a multivariate regression model was used to identify factors associated with visual acuity enhancement (at least one line on Snellen charts). Low initial visual acuity and older age were strong negative prognostic factors (p<0.001 for both). Midperipheral ulcer location was a significant positive factor for logMAR reduction (p=0.049). Table 7 depicts the analysed factors in an ANCOVA model.
Table 7. Multivariate regression model (ANCOVA model) for logMAR improvement in the whole cohort of patients (n=497).
Variable |
Estimate |
(95% CI) |
p-value |
Intercept |
0.029 |
-0.381; 0.439 |
0.890 |
Distance Visual acuity - admission |
-0.593 |
-0.730; -0.455 |
<0.001 |
Age (years) |
0.013 |
0.008; 0.018 |
<0.001 |
Sex: Male |
0.193 |
-0.014; 0.400 |
0.067 |
Ulcer location: Midperiphery* |
-0.241 |
-0.481; -0.001 |
0.049 |
Glaucoma |
0.247 |
-0.070; 0.563 |
0.126 |
Corneal transplant (therapeutic) |
-0.278 |
-0.566; 0.011 |
0.059 |
Polytherapy |
-1.043 |
-2.207; 0.121 |
0.079 |
Antifungal |
-0.187 |
-0.410; 0.035 |
0.098 |
Post-keratoplasty infectious keratitis |
0.285 |
-0.028; 0.598 |
0.074 |
bold indicates statistically significant p-values
Discussion
Our study demonstrates that inpatient treatment significantly improves visual acuity in severe keratitis affecting both virgin and post-keratoplasty corneas, though outcomes were notably worse in the post-keratoplasty microbial keratitis group. BCVA improved by at least one line in 52% of patients with naïve keratitis and 33% of those with PKMK. Median BCVA on admission was 1.9 logMAR (counting fingers) for virgin corneas and 2.3 logMAR (hand movement) for PKMK, improving to 1.3 logMAR (2.5/50 Snellen chart) and 1.9 logMAR (counting fingers), respectively, at discharge. Legal blindness status improved in 14% of patients with naïve keratitis but only in 5% of the PKMK group. Near visual acuity was detectable in 27% of patients at admission and in 35% at discharge, with advancement seen in 27% of the virgin cornea group and just 8% in the PKMK group (p < 0.001).
Legal blindness status remains in 67% patients of the entire cohort at discharge.
Low initial VA and older age are widely recognized as unfavorable prognostic factors for microbial keratitis treatment [11-15]. Few studies, however, have measured VA improvements following intensive inpatient treatment or directly compared outcomes between severe keratitis in virgin and transplanted corneas within a single cohort. Unlike prior research, our study included all visual acuity levels, including hand movement (2.3 logMAR), light perception (2.7 logMAR), and no light perception (3.0 logMAR) [16].
Studies on PKMK generally show poorer than naïve cornea keratitis visual prognosis. For instance, Chatterjee reported logMAR < 1.0 in only 21% of post-treatment PKMK patients [17], while Ong documented mean logMAR of 1.69 at admission and Ittah-Cohen noted a shift from 1.7 to 0.98 logMAR after treatment [18,19]. Better outcomes have been reported in select cases: Atta observed improvement from 0.98 to 0.44 logMAR in culture- and PCR-negative ulcers, while Cabrera-Aguas reported a VA improvement from 1.7 to 0.98 logMAR in a mixed cohort [4,12,20]. Saeed showed a shift from 0.76 to 0.24 logMAR in contact lens (CL)-wearers [21]. Culture-negative cases were similarly linked to favorable outcomes in Keay's study [22]. A predictive model for VA outcomes in microbial keratitis identified older age, low initial VA, and corneal transplant history as negative factors, aligning with our findings [15].
Our cohort represents one of the largest post-keratoplasty infectious keratitis groups, directly compared to severe keratitis in virgin corneas. Differences include lower prevalence of CL use and ocular trauma compared to other studies [23,24], high rates of mixed infections (30% vs. ~10% in other large cohorts) [29,30], high glaucoma prevalence, affecting 39% of PKMK and 12% of naïve keratitis patients, likely impairing healing due to long-term antiglaucoma medication use [25].
The longer median hospitalization time for PKMK cases (9 vs. 8 days in the virgin cornea group) reflects general trends [14]. Hospital stay correlated with VA improvement, but no statistical significance was seen in multivariate regression analysis. Seasonal variation showed more severe cases in spring and summer, consistent with prior findings, though final VA was not season-dependent [26,27].
Gram-positive bacteria dominated (52%), consistent with prior research [28]. Notably, our cohort had a high rate of mixed infections (30%), comparable only to Wong's New Zealand study [30]. Gram-negative bacteria were less common (16%), with PKMK cases showing even lower rates (8% vs. 17% in virgin corneas), reflecting the limited number of CL wearers [21,26,31]. Interestingly, Gram-negative pathogens like Pseudomonas aeruginosa have been linked to better final VA outcomes [16,32]. Thus, both high prevalence of mixed infection combined with low incidence of CL wear might contribute to our unfavourable visual results.
There is no universal guideline for microbial keratitis treatment [33,34]. A global survey reported that 20% of ophthalmologists prefer monotherapy (fluoroquinolones), while 78% opt for polytherapy, typically combining aminoglycosides with beta-lactams or vancomycin [35]. Our study found no significant difference in final VA between mono- and polytherapy, except in the PKMK group, where polytherapy led to better outcomes. Nayel demonstrated superior healing rates with gentamicin plus vancomycin compared to moxifloxacin monotherapy or ceftazidime-vancomycin combinations [37]. Given the high prevalence of Gram-positive bacteria and increasing resistance to antibiotics (mainly Methicillin-resistant Staphylococcus aureus), our approach of using vancomycin plus aminoglycoside remains optimal [37-40].
In summary, best corrected visual acuity got better of at least one Snellen line in 47% of patients during hospital admission due to corneal ulcer. However, the discrepancy between naïve cornea and transplanted cornea was significant (52% vs 33%, respectively). Median BCVA improved from 1.9 logMAR to 1.5 logMAR for the entire cohort but the discrepancy between the groups was observed. Younger age, better initial VA and corneal midperiphery location represent positive prognostic factors for the visual outcome. Finally, 67% of patients with corneal ulcer admitted to the hospital are discharged with legal blindness level of BCVA. Near visual acuity enhanced in 23% of patients.
The main limitation of this study is the lack of knowledge about baseline visual acuity prior to the infection onset, particularly in transplanted corneas, which might contribute to the initially low VA. Preexisting blindness cannot be excluded. The retrospective nature of the project limited our ability to obtain all corneal culture results (especially from years 2008-2015 when solely paper print records from the microbiology department were provided). Furthermore, themajority of our patients were referred to our tertiary centre as non-healing ulcers with empirical treatment. Thus, corneal cultures were not obtained due to ongoing extensive antimicrobial therapy. Lastly, as our study design is limited to the hospitalisation period, we were unable to assess final VA at the time of ultimately healed ulcers.
Considering the global burden of complications and the occupational and psychological impact on patients’ lives, it is of utmost importance to improve keratitis care. For proper counselling, we need to provide data regarding the level of improvement after hospital admission treatment, including near vision. This study presents one of the largest cohorts of microbial keratitis patients from Central-Eastern Europe, where there is a scarcity of epidemiological studies on corneal ulcers, contributing to the local and ethnic characteristics of several ocular disorders.”
Comments 10: The English in the present manuscript is not of publication quality and require some improvement and proper proof-reading.
Response 10: Thank you for that comment. Professional English proof-reading has been performed. We hope it responds to the expectations.
We again would like to thank you for your valuable feedback and hope that you accept our incorporation of the comments into our amended manuscript.
Sincerely,
Joanna Przybek-Skrzypecka, MD, PhD
Corresponding author

Round 2
Reviewer 2 Report
Comments and Suggestions for Authors
Dear authors,
I highly appreciate your efforts in addressing my comments and suggestions point by point, and I am pleased with how your final revisions have been incorporated into the manuscript. So far, everything looks good, and I have no concerns with the revised version